# A Focused Review on the Maximal Exercise Responses in Hypo- and Normobaric Hypoxia: Divergent Oxygen Uptake and Ventilation Responses

**DOI:** 10.3390/ijerph17145239

**Published:** 2020-07-20

**Authors:** Benedikt Treml, Hannes Gatterer, Johannes Burtscher, Axel Kleinsasser, Martin Burtscher

**Affiliations:** 1Department of General and Surgical Intensive Care, Medical University Innsbruck, 6020 Innsbruck, Austria; benedikt.treml@tirol-kliniken.at; 2Institute of Mountain Emergency Medicine, Eurac Research, 39100 Bolzano, Italy; hannes.gatterer@eurac.edu; 3Institute of Sport Sciences, Synathlon, Uni-Centre, 1015 Lausanne, Switzerland; johannes.burtscher@unil.ch; 4Postoperative Critical Care Unit, Department of Anesthesiology and Critical Care Medicine, Medical University Innsbruck, 6020 Innsbruck, Austria; 5Department of Sport Science, University of Innsbruck, 6020 Innsbruck, Austria; martin.burtscher@uibk.ac.at; 6Austrian Society for Alpine and Mountain Medicine, 6020 Innsbruck, Austria

**Keywords:** exercise testing, normobaric, hypobaric, maximal oxygen uptake, maximal minute ventilation

## Abstract

The literature suggests that acute hypobaric (HH) and normobaric (NH) hypoxia exposure elicits different physiological responses. Only limited information is available on whether maximal cardiorespiratory exercise test outcomes, performed on either the treadmill or the cycle ergometer, are affected differently by NH and HH. A focused literature review was performed to identify relevant studies reporting cardiorespiratory responses in well-trained male athletes (individuals with a maximal oxygen uptake, VO_2*max*_ > 50 mL/min/kg at sea level) to cycling or treadmill running in simulated acute HH or NH. Twenty-one studies were selected. The exercise tests in these studies were performed in HH (*n* = 90) or NH (*n* = 151) conditions, on a bicycle ergometer (*n* = 178) or on a treadmill (*n* = 63). Altitudes (simulated and terrestrial) varied between 2182 and 5400 m. Analyses (based on weighted group means) revealed that the decline in VO_2max_ per 1000 m gain in altitude was more pronounced in acute NH vs. HH (−7.0 ± 1.4% vs. −5.6 ± 0.9%). Maximal minute ventilation (VE*_max_*) increased in acute HH but decreased in NH with increasing simulated altitude (+1.9 ± 0.9% vs. −1.4 ± 1.8% per 1000 m gain in altitude). Treadmill running in HH caused larger decreases in arterial oxygen saturation and heart rate than ergometer cycling in acute HH, which was not the case in NH. These results indicate distinct differences between maximal cardiorespiratory responses to cycling and treadmill running in acute NH or HH. Such differences should be considered when interpreting exercise test results and/or monitoring athletic training.

## 1. Introduction

When Mexico City (2300 m above sea level) was chosen to host the 1968 Olympic Games, sports scientists and coaches were challenged with performance limitations and physiological responses to exercise in acute hypobaric hypoxia (HH) [1,2,3]. Subsequently, there was a rise in scientific interest in evaluating the effects of acute normobaric hypoxia (NH) on exercise performance and on physiological responses [4,5]. Today, some 50 years later, comparisons between acute HH and NH still remain scarce [6,7,8]. It has to be noted that during the acclimatization process (with subacute and chronic exposures to altitude/hypoxia) physiological responses change associated with improved exercise performance [9,10].

A few years ago, Coppel and coworkers systematically reviewed the literature for physiological response differences between HH and NH [11]. This review and a more recent study [12] mainly focused on resting and submaximal exercise responses, with less emphasis on maximal exercise outcomes. Interestingly, most results indicated small differences in ventilatory parameters, however contrasting reports also exist [6,7,8]. Such divergent findings come as little surprise, as comparisons have often been affected by confounders such as the duration spent in hypoxia, temperature, and the humidity during exposure to hypoxia. Moreover, in some studies small sample sizes provide only limited statistical power [11,13,14].

With regard to potential exercise performance differences between HH and NH, the few existing comparative studies suggest a larger performance decline (e.g., time trial performance) in HH compared to NH [15]. Yet, little information is available on differences in physiological responses of athletes performing maximal exercise testing (i.e., incremental exercise test to exhaustion) in acute HH or NH.

Therefore, the aim of the present review was to summarize selected findings of studies investigating maximal incremental exercise test outcomes of well-trained athletes acutely exposed to HH (simulated and terrestrial) or NH.

## 2. Materials and Methods

A focused literature review was performed to identify relevant studies reporting cardiorespiratory responses of athletes performing cycling or treadmill running in acute HH or NH. The literature search was performed in the following databases using a cut-off date of May 2019: Pubmed/Medline, Web of Science, Science direct, Scopus, and Sport Discus. The following keywords were used: VO_2max_ and acute hypoxia (or altitude) and athletes. Figure 1 displays a flow chart which depicts the selection process.

Inclusion criteria were well-trained male athletes (due to insufficient data on females), not exposed to altitudes above 2000 m for at least one month prior to the study, with a maximal oxygen consumption (VO_2max_ ≥ 50 mL/min/kg) at sea level and studies reporting the outcomes of maximal incremental exercise tests (cycling or treadmill) completed at sea level and after acute exposure (ie., within the first 2 h, except one study where the ascent to real altitude lasted 8 h [9]) to HH (simulated and terrestrial) or NH at an altitude greater than 2000 m. Exercise test outcomes included VO_2max_, maximal minute ventilation (VE_max_), peripheral oxygen saturation at maximal exercise (SpO_2_), and maximal heart rate (HR_max_).

### Statistics

Data are presented as weighted group means and standard deviation (SD) of the mean. To allow for comparisons between different altitudes, changes in cardiorespiratory variables from sea level to altitude were calculated as deltas per 1000 m gain in altitude. The association between continuous variables was assessed using simple linear regression analyses. Potential differences of cardiorespiratory responses to HH or NH during cycling or treadmill running were evaluated by calculation of effect sizes (ES, *Cohen’s d*). An ES below 0.5 was considered to be small, 0.5 to 0.8 to be medium, and >0.8 to be large [16].

## 3. Results

A total of 21 studies were included, with 26 groups (*n* = 241 participants) evaluated at sea level and altitude (i.e., in NH or HH corresponding to altitude levels varying between 2182 and 5400 m). Out of 26 groups, 10 (*n* = 90) were performed in HH [9,17,18,19,20,21,22,23] (all in hypobaric chambers, except one which was performed at real altitude [9]), and 16 (*n* = 151) in NH [24,25,26,27,28,29,30,31,32,33,34]. Baseline characteristics of the groups included and the general physiological responses to maximal exercise testing at altitude are shown in Table 1.

Figure 2 depicts the salient findings of this focused review in regard to divergent oxygen uptake and ventilation responses in hypo- and normobaric hypoxia.

Effect sizes for differences between cycling and treadmill running could not be calculated because there was only one study evaluating treadmill running.

Table 2 displays the characteristics of the study participants of the studies included, separated by the HH and NH condition.

Table 3 depicts the differences between HH and NH per 1000 m of altitude gain separated by exercise task.

The data show that in NH, VO_2max_, combined for cycling and treadmill running, is reduced to a larger extent compared to HH (ES: 1.19). VE_max_, combined for cycling and treadmill running, is slightly increased in HH, whereas it is reduced in NH (ES: 2.32); small to medium ES were detected for SpO_2_ (ES: 0.27) and HR_max_ (ES: 0.49) between conditions. Interestingly, when converting VE_max_ from body temperature pressure saturated (BTPS) to standard temperature pressure dry (STPD), VE_max_ values (STPD) were the same in NH and HH. Moderate to large effects were found between cycling and treadmill running in HH, indicating that VO_2max_ (ES: 0.74), SpO_2_ (ES: 0.94) and HR_max_ (ES: 0.66) are more affected during treadmill running compared to cycling, although VE_max_ was slightly higher during treadmill running (ES: 0.35).

Furthermore, the data show a similar slope between changes in VO_2max_ and SpO_2max_ (Figure 3) and VE_max_ and HR_max_ (Figure 4) with gain in altitude, but at different levels for NH and HH.

## 4. Discussion

The main outcomes of the present literature review are that (for male athletes) the VO_2*max*_ decline is larger in NH as compared to HH (simulated and terrestrial) and that VE*_max_* slightly increases in HH whereas it is reduced in NH. Additionally, for each 1000 m of altitude gain in HH (simulated and terrestrial), VO_2*max*_ during treadmill running seems to be more depressed compared to cycling. Furthermore, during treadmill running in HH (simulated and terrestrial), when compared to cycling, a larger SpO_2*max*_ and HR*_max_* decrease per 1000 m of altitude gain was found.

The VO_2*max*_ reduction and the increase of VE*_max_* at altitude are well documented [1,25,35,36,37]. It is also known that VO_2*max*_ is reduced in both acute NH [25,27,28,29,30,32,33,34,38,39] and HH [9,17,18,19,20,21,22,23] conditions. Yet, information on specific VO_2*max*_ differences between acute NH and HH in trained individuals is still lacking.

Previous work comparing differences between cardiorespiratory responses in acute HH and NH focused primarily on resting and submaximal exercise conditions [6,7,8,11,12,40]. Results of a recent meta-analysis (primarily performed on studies investigating resting conditions) indicate differences in variables related to VE, NO, fluid retention, and acute mountain sickness (AMS) associated factors between acute HH and NH [11]. The existence of such differences is also supported by Saugy et al. and Beidleman et al. who found differences in time trial performance between acute HH and NH, with greater performance losses during acute HH exposure [15,41]. In the study of Saugy et al. this was thought to be a result of a lower SpO_2_ in acute HH compared to NH, which was not reported by Beidleman et al. Both studies, however, did not determine cardiorespiratory responses at maximal exercise, which is of particular interest from an athletic training and testing perspective. To bridge this knowledge gap, the present analyses focused primarily on differences between acute NH and HH at maximal exercise. Larger VE*_max_* responses in acute HH compared to NH (at comparable simulated altitude levels) may be expected due to the lower air density in HH [35]. Higher VE*_max_* in acute HH compared to NH (during running) was recently demonstrated by Ogawa et al., which is likely due to a lower flow resistance in the airways [42]. When linking the reduced VE*_max_* responses to the higher VO_2*max*_ decline in NH compared to HH, the lower VE*_max_* at reduced exercise performance has to be emphasized. This complicates the interpretation of VE*_max_* changes when exercise is not performed at the same level of hypoxia in acute HH and NH. Distinct breathing patterns (i.e., tidal volume and breathing frequency) in acute HH and NH [43,44,45] might have influenced the central motor drive [46] and VO_2*max*_. Irrespective of these proposed mechanisms, it should be mentioned that Pugh demonstrated 50 years ago that exercise VE expressed as STPD remained unchanged at various altitudes at least up to 5800 m, while exercise VE expressed as BTPS (as is usually done) increased with the gain in altitude in a non-linear fashion [35]. Accordingly, similar VE*_max_* values (STPD) in HH and NH were found when converting the VE_max_ values of the included studies from BTPS to STPD.

It is worth noting that a similar decrease of SpO_2_ at maximal exercise—despite different VE*_max_* responses—was related to a more reduced VO_2*max*_ in acute NH than in HH. Arterial oxygen desaturation has been shown to be linked to the decrease in HR*_max_*. Decreases in HR_max_, moreover might be influenced by the activity of the parasympathetic system or by myocardial electrophysiological and central factors [47,48,49]. Analyses of the results of the studies included revealed a similar relationship in acute NH (*r*^2^ = 0.8, *p* < 0.05) but not in HH. Whether this association is actually more important in acute NH than in HH has yet to be established. A higher SpO_2_ might be related to a higher performance level. Overall, it seems that there is a complex interplay between different respiratory, metabolic and autonomic regulations in NH and HH that might explain the observed differences in VO_2*max*_.

Beside the general responses of performing exercise in acute HH and NH, we found some small differences in the maximal cardiorespiratory responses between cycling and treadmill running. These differences are not straightforward to explain. Gavin and Stager demonstrated a more pronounced exercise-induced hypoxaemia (in normoxia) during cycling compared to running, potentially indicating a similar association in HH [50]. Generally, breathing patterns are different between cycling and running [51], potentially explaining the differences observed. Moreover, there is one excellent review specifically dealing with physiological differences between cycling and running [52]. Although differences will largely depend on the sport and training practiced, different responses to maximal exercise performed in normoxia, as outlined in that review [52], may help to explain such differences in hypoxia, as demonstrated in the present review. Millet and colleagues reported the following differences [52]. Whereas runners usually achieve higher VO_2*max*_ values, in cyclists those values seem not to be different between cycling and treadmill running. Maximal heart rates have been found to be about 5% higher when derived from incremental treadmill running compared to cycling (at least in not well-trained individuals). Moreover, the delta efficiency was shown to be higher in running. Exercise ventilation is likely influenced by different pulmonary mechanics between cycling and treadmill running and may be more impaired in cycling than running. With regard to desaturation during maximal exercise, triathletes at least showed more pronounced desaturation (lower SpO_2_ values) during running than cycling.

In the present literature review, the focus was on maximal values during exercise testing. These results to some extent contrast those found during submaximal exercise. For instance, Netzer et al. found lower SpO_2_ and a higher HR values during exercise in HH compared to NH [53]. Similar to the present findings, Faiss et al. reported no differences in SpO_2_ values between conditions during cycling, yet VE was lower in acute HH compared to NH [45]. Similar to this, Beidleman et al. and Basualto-Alarcón et al. reported no effect of the type of condition on SpO_2_ and HR [6,15], yet exercise performance was more affected under acute HH conditions [15]. The differences in VE between submaximal and maximal exercise in acute NH and HH seem reasonable; the lower air density may allow a lower VE at submaximal exercise and a higher one at maximal exercise. With regard to exercise performance and SpO_2_, the factors discussed above (e.g., metabolic and autonomic regulation) may explain some of the differences, yet this has to be established in future studies.

### Limitations

Some limitations of the present analyses have to be acknowledged. First, we are aware that there have been a multitude of studies performing maximal incremental exercise testing in acute hypoxia. Yet, as the focus of those studies was on other topics, our literature search strategy might not have identified them. Nonetheless, the number of included studies and participants seems acceptable to allow the preliminary conclusions here presented. Second, analyses of data in this literature review have been performed on the basis of group averages, allowing effect sizes to be calculated. Third, we were not able to evaluate potential gender differences, as there are only very few studies including female athletes. Fourth, we must acknowledge that the validity of any comparison between acute NH and HH conditions depends on the accuracy of the calculation of the equivalent altitude, as was recently addressed in detail [40,54]. The calculation may depend on the location of the experimental site, the ambient temperature, or the degree of hygrometry and the official standard used [54]. In the present analysis, these factors were disregarded. Nonetheless, from a practical/real life perspective, our analysis seems valid, yet we urge others to perform studies designed to specifically address this issue (e.g., cross over studies with matched PiO_2_). In addition, available information does not allow whether participants were blinded in NH or not to be assessed with confidence. Thus, a potential placebo effect cannot be excluded. Finally, potential confounding may arise from the use of different test protocols and different measurement devices. Overall, this is the first review taking into account a reasonable number of studies evaluating cardiorespiratory exercise responses to maximal incremental exercise testing (treadmill and cycle ergometry) in trained individuals at sea level and in acute HH (simulated and terrestrial) and NH as well.

## 5. Conclusions

The analyses presented indicate that in well-trained men performing maximal exercise tasks, VO_2*max*_ declines to a larger extent in acute NH compared to HH, which goes along a higher VE*_max_* in HH. Treadmill running in acute HH caused more pronounced depressions of VO_2*max*_, SpO_2_, and HR*_max_* compared to cycling exercise. These findings may have practical and clinical implications when interpreting exercise test results in acute NH and HH, and could inspire the design of according studies, but are also of novel applications in exercising in hypoxia.

Future studies will focus on potential sex and age differences between responses to acute NH and HH, but will also evaluate changes occurring with acclimatization.

## Figures and Tables

**Figure 1 ijerph-17-05239-f001:**
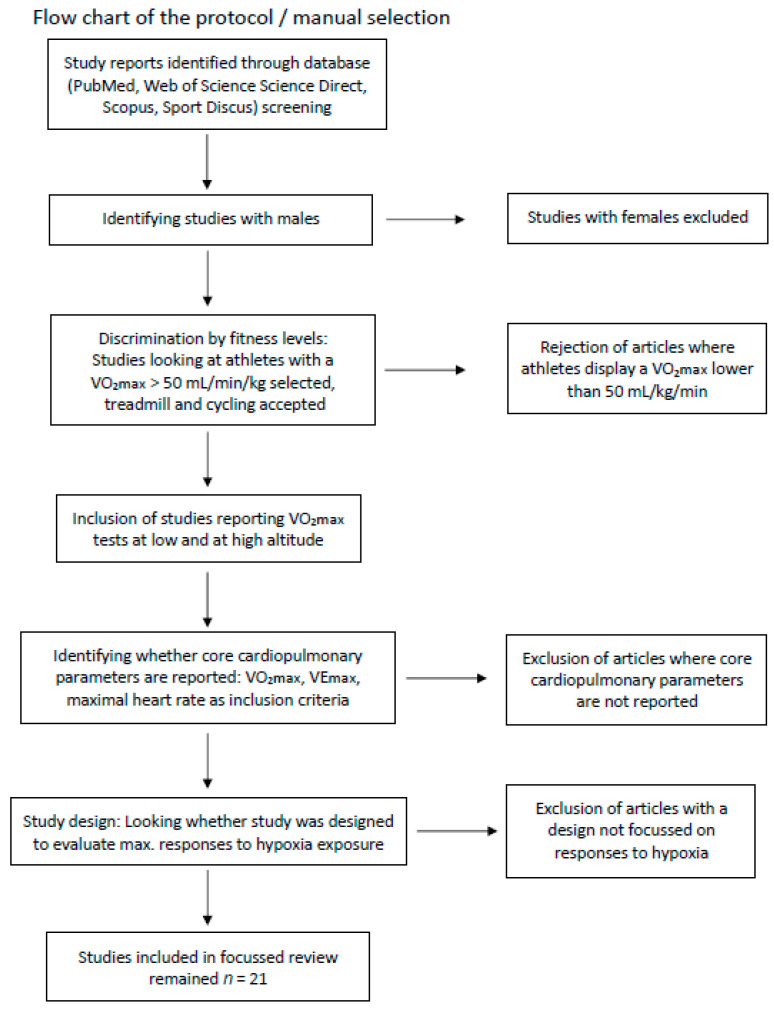
Flow chart of the protocol.

**Figure 2 ijerph-17-05239-f002:**
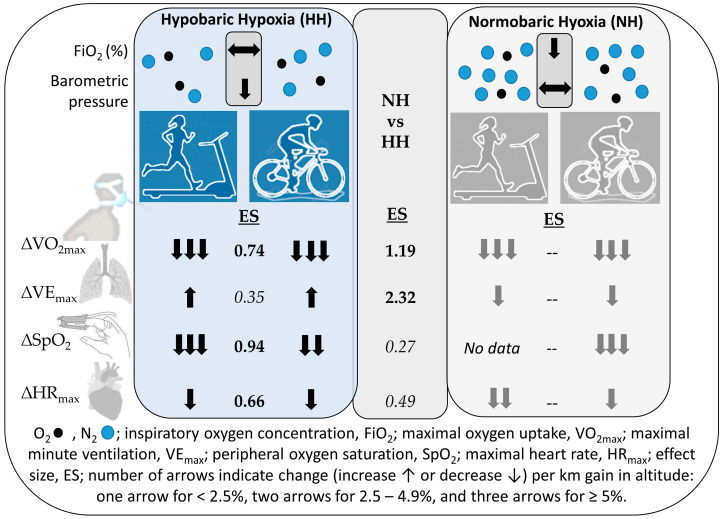
Schematic presentation of the key findings.

**Figure 3 ijerph-17-05239-f003:**
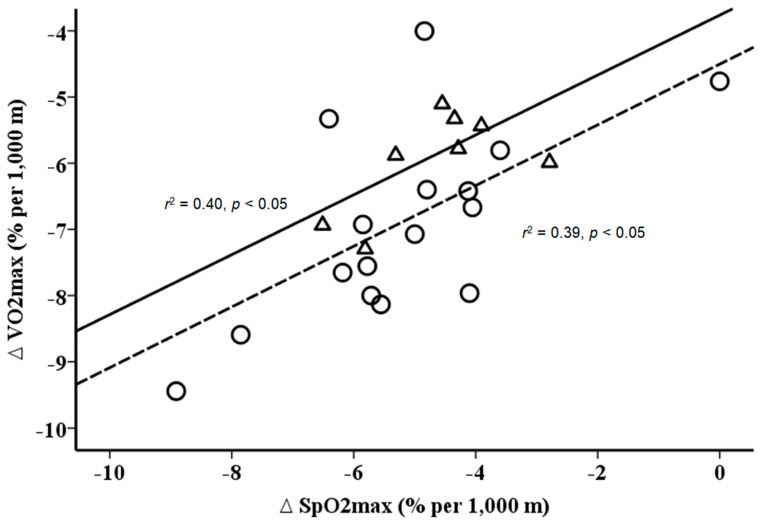
Relationship between changes (%) in peripheral oxygen saturation during maximal exercise (SpO_2max_) and maximal oxygen uptake (VO_2max_) per 1 km gain in altitude. Solid line (triangles) for hypobaric hypoxia (*r*^2^ = 0.40, *p* < 0.05) and dashed line (circles) for normobaric hypoxia (*r*^2^ = 0.39, *p* < 0.05). Delta values are calculated as HH/NH values minus sea level values.

**Figure 4 ijerph-17-05239-f004:**
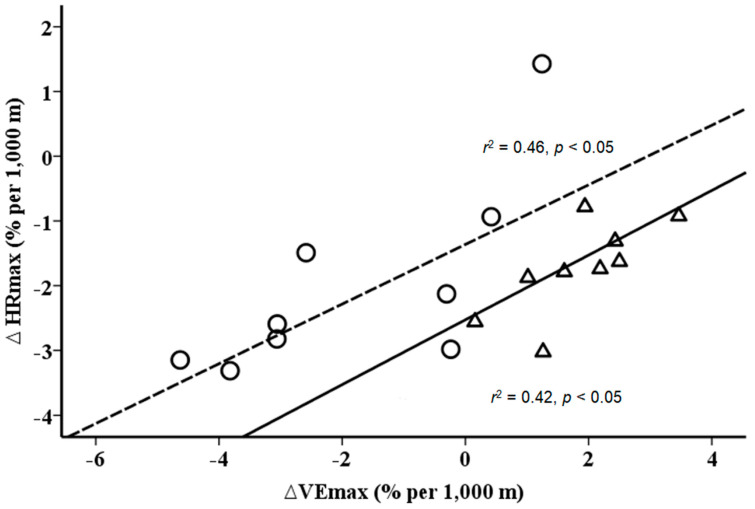
Relationship between changes (%) in maximal minute ventilation (VE_max_) and maximal heart rates (HR_max_) per 1 km gain in altitude. Solid line (triangles) for hypobaric hypoxia (*r*^2^ = 0.46, *p* < 0.05) and dashed line (circles) for normobaric hypoxia (*r*^2^ = 0.42, *p* < 0.05). Delta values are calculated as HH/NH values minus sea level values.

**Table 1 ijerph-17-05239-t001:** Characteristics of the studies included in the analysis.

Reference	Participants	VO_2max_ at Sea Level	Altitude or Equivalent	Hypoxia Condition	Baro-Metric Pressure	F_i_O_2_	Exercise Mode	VO_2max_ at Altitude
	(number)	(mL/min/kg)	(m)		(mmHg)	(%)		(mL/min/kg)
Beidleman et al. (1997) [17]	6	57.0	4300	HH (HC)	446		treadmill	40.0
Benoit et al. (2003) [24]	12	64.2	5400	NH		10.4	cycling	36.0
Benoit et al. (2003) [24]	17	50.8	5400	NH		10.4	cycling	31.4
Fagraeus et al. (1973) [18]	11	50.1	3250	HH (HC)	520		cycling	43.5
Ferretti et al. (1997) [25]	5	62.1	2182	NH		16.0	cycling	53.4
Ferretti et al. (1997) [25]	5	62.1	4966	NH		11.0	cycling	35.6
Friedmann et al. (2007) [26]	20	68.0	2682	NH		15.0	treadmill	53.1
Fulco et al. (1988) [19]	7	50.1	4300	HH (HC)	446		cycling	37.2
Gavin et al. (1998) [27]	7	60.4	3591	NH		13.3	cycling	43.8
Gavin et al. (1998) [27]	6	63.7	3591	NH		13.3	cycling	42.1
Heubert et al. (2005) [28]	9	62.7	2500	NH		16.0	cycling	53.6
Horstman et al. (1980) [9]	9	51.0	4300	HH (TA)	460		treadmill	35.0
Koistinen et al. (1995) [20]	12	57.4	3200	HH (HC)	520		cycling	46.6
Lawler et al. (1988) [29]	7	65.0	3207	NH		14.0	cycling	51.1
Martin et al. (1993) [30]	8	67.2	3760	NH		13.0	cycling	49.7
Mollard et al. (2007) [31]	8	65.5	2479	NH		15.4	cycling	57.7
Mollard et al. (2007) [31]	8	65.5	4527	NH		11.7	cycling	46.4
Ofner et al. (2014) [32]	10	51.0	3500	NH		14.0	cycling	42.5
Peltonen et al. (2001) [33]	6	61.8	2682	NH		15.0	cycling	48.6
Robach et al. (2008) [34]	7	50.0	2500	NH		15.3	cycling	42.0
Robach et al. (2008) [34]	7	50.0	3500	NH		13.4	cycling	36.0
Robach et al. (2008) [34]	7	50.0	4500	NH		11.5	cycling	33.0
Robergs et al. (1998) [23]	14	60.4	2559	HH (HC)	566		cycling	52.0
Squires et al. (1982) [21]	12	60.0	2286	HH (HC)	574		treadmill	53.0
Wehrlin et al. (2006) [22]	8	66.1	2300	HH (HC)	573		treadmill	58.0
Wehrlin et al. (2006) [22]	8	66.1	2800	HH (HC)	573		treadmill	55.4

Maximal oxygen uptake, VO_2*max*_; fraction of inspired oxygen, F_i_O_2_; hypobaric hypoxia, HH (HC, hypobaric chamber; TA, terrestrial altitude); normobaric hypoxia, NH.

**Table 2 ijerph-17-05239-t002:** Characteristics of study participants of the included HH and NH studies.

		All		Divided by Exercise Mode			
**Hypobaric Hypoxia Studies**							
**Variable**	***n***			***n***	**Cycling Exercise**	***n***	**Treadmill Running**
Age (yr)	90	25.7 ± 3.7		47	26.1 ± 4.3	43	25.3 ± 3.0
Weight (kg)	90	73.6 ± 2.5		47	73.5 ± 1.2	43	73.7 ± 3.4
Height (cm)	90	178.8 ± 2.0		47	178.1 ± 2.1	43	179.5 ± 1.5
Altitude level (m)	90	3117 ± 755		47	3144 ± 577	43	3087 ± 918
**Normobaric Hypoxia Studies**							
Age (yr)	147	25.2 ± 3.4		129	26.0 ± 2.9	18	24.1 ± 4.1 *
Weight (kg)	147	72.6 ± 5.2		129	72.9 ± 5.5	18	71.2 ± 6.5
Height (cm)	147	178.7 ± 2.7		129	178.2 ± 2.5	18	182.1 ± 5.9
Altitude level (m)	147	3678 ± 1095		129	3830 ± 1099	18	2682

Maximal oxygen uptake, VO_2max_; oxygen saturation, SpO_2_; maximal minute ventilation, VE_max_; gas volumina are expressed as body temperature pressure saturated (BTPS). * Standard deviation (SD) taken from the respective study.

**Table 3 ijerph-17-05239-t003:** Changes for each 1000 m altitude climb separately shown for altitude conditions and exercise mode.

	HH		NH		HH				NH			
Variable	*n*	Cycling + Treadmill	*n*	Cycling + Treadmill	*n*	Cycling Exercise	*n*	Treadmill Running	*n*	Cycling Exercise	*n*	Treadmill Running
∆VO_2max_ (%/km)	9 90	−5.6 ± 0.9	18 151	−7.0 ± 1.4 ^a^	4 47	−5.3 ± 0.7	5 43	−5.9 ± 0.9 ^b^	17 131	−6.8 ± 1.4	1 20	−8.1 ± 0.0
∆VE_max_ (%/km)	9 90	1.9 ± 0.9	15 132	−1.4 ± 1.8 ^a^	4 47	1.8 ± 1.1	5 43	2.1 ± 0.5 ^c^	14 112	−1.6 ± 1.9	1 20	−0.2 ± 0.0
∆SpO_2_ (%/km)	8 79	−4.6 ± 1.0	17 131	−5.0 ± 1.8 ^c^	3 36	−4.2 ± 0.9	5 43	−5.0 ± 0.8 ^a^	17 131	−5.0 ± 1.9	/	/
∆HR_max_ (%/km)	9 90	−1.6 ± 0.6	13 117	−2.1 ± 1.3 ^c^	4 47	−1.4 ± 0.7	5 43	−1.8 ± 0.5 ^b^	12 97	−2.0 ± 1.3	1 20	−3.0 ± 0.0

Heart rate, HR; hypobaric hypoxia, HH; maximal oxygen uptake, VO_2max_; normobaric hypoxia, NH; oxygen saturation, SpO_2_; ventilation, VE. Delta values are calculated as HH/NH values minus sea level values. *n* indicates the number of included studies (upper number) and the number of participants involved (lower number). ^a^, ^b^ and ^c^ indicate large, medium, and small effect size respectively. ES for treadmill running in NH could not be calculated because only one study was available.

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
