# Peer review of "A Focused Review on the Maximal Exercise Responses in Hypo- and Normobaric Hypoxia: Divergent Oxygen Uptake and Ventilation Responses"

_ijerph, 2020, doi:10.3390/ijerph17145239_

Round 1
Reviewer 1 Report
This is a comprehensive review of studies addressing differences between normobaric and hypobaric hypoxia on exercise performance in highly trained subjects.
This is in line with numerous studies aiming to evidence a difference between these two conditions, without convincing !
The main problem in comparing these two conditions is that the equivalence between the inspired PO2 is not controlled, either because the FIO2 or the barometric pressure is not correctly monitored. A slight difference in FIO2 can lead to a great difference in “equivalent” altitude so that a majority of studies are not comparable. Sometimes, also, authors do not take into account the pressure of water vapor in the computation of equivalent altitude.
In the present review, it is not stated if selected studies correctly monitored these fundamental variables. Inspired PIO2 should be mentioned.
Another substantial limitation of this review is that most studies explored either HH or NH but not HH and NH in the same subjects, knowing that the best methodological approach would be a cross-over study.
Specific questions:
The reviewer does not understand on what arguments the authors give their “Key findings of Table 2” ?
Tables 3a and 2b, what is the “altitude”… all studies are done at various altitudes: this is not clear for the reviewer.
Line 107: 8 instead of (
The reviewer supposes that all studies are done in “acute” conditions: it is easy to conceive for NH but for HH, it needs a certain time for acclimatization, especially for high altitudes. This may interfere with the values of ventilation, SpO2, VO2max, etc.
From a mechanistic point of view, it is clear and well known that the decrease in air density and therefore in work of breathing may play a major role in the small observed differences.
Figures 1 and 2: where are the statistics showing that the lines are parallel but significantly shifted ?
Please change SaO2 for SpO2 everywhere (in figures).
Concerning max HR (lines 177-185), the statements made by the authors are not clear and somewhere erroneous. HR (or more exactly cardiac output) may have an influence on SpO2 since increasing HR increases the diffusion limitation in the lungs (decreased lung circulation time) but decreased SpO2 (hypoxia in general) decreases maximal HR but TWO mechanisms (the authors quote Mourot’s paper but this is a partial view of the problem): both increased parasympathetic activity and desensitization of adrenergic system lead to a decrease in maxHR, but this decrease is only seen after 2 or 3 days at altitude, not in acute hypoxia. Other mechanisms may interfere in very acute hypoxia.
Line 213: “considerable” is not appropriate !
Considering the difference between treadmill and cycling, this paper does not bring new information. Moreover, in field conditions, running and cycling maximal performances may highly depend on air resistance, which is not the case in laboratory conditions.
The authors may have missed a recent Cross Talk in the Journal of Physiology that would have been interesting to quote in your discussion….
Millet GP & Debevec T (2020). CrossTalk proposal: Barometric pressure, independent of PO2, is the forgotten parameter in altitude physiology and mountain medicine. J Physiol 598, 893-896.
Richalet JP (2020). CrossTalk opposing view: Barometric pressure, independent of PO2, is not the forgotten parameter in altitude physiology and mountain medicine. J Physiol 598, 897-899.
Author Response
Re: Thank you very much. This issue is comprehensively reviewed in the CrossTalk article of Richalet (2020) and Millet et al. (2020). We added this issue to the limitation section and included the CrossTalk articles in the reference list (see page 11, line 238ff).
We also would like to add that in the present analysis we adopted a somewhat unusual approach to compare NH and HH effects. To the best of our knowledge is no study available. As most studies dealt with submaximal exercise like time trails, that directly compared incremental maximal exercises test results performed in HH and NH we chose to analyze studies performing such tests in normoxia and then either in HH or NH. We analyzed the performance decline from different studies in both conditions and compared the results. Obviously, this will not give a conclusive result (because of the issues you mentioned), but from a practical/real life point of view it may represent a valid analysis that should be addressed in specifically designed research projects.
Re: We fully agree on that. Unfortunately, (to the best of our knowledge) there are no studies that follow this approach up to now. Therefore, we followed this less robust approach as outlined before. We addressed this issue in the limitation section (see comment before).
Re: We decided to replace table 2 by a “didactic figure” (now figure 2, see page 6)
Re: The reviewer is right; this may be a bit misleading. We decided to solely present the characteristics of the participants in one Table 2 without panel a and b (former Table 3, see page 6). Changes per 1000 m of altitude are depicted in Table 3 (former Table 4, see page 8), which is, in our view, a better way to present our findings considering the way we did the analyses.
Re: Changed, thank you.
Re: All HH studies, except one have been performed in hypobaric chambers. The exposure duration before starting the test was between 2 min and 2 h in the HH chamber setting. In the only study performed at real altitude (Horstmann et al., 1980), the time to reach the altitude level of testing was 8 h. We did not exclude this study because it may be interesting for comparison reasons. It did not change the presented findings when excluded. We included the information on the settings in Table 1 (see page 4f). Additionally, we included the following in the limitation section “Finally, potential confounding may arise from the use of different test protocols, exposure duration prior to testing and different measurement devices.” (see page 12, line 260ff).
Re: We agree, and are confident that we addressed this adequately in the manuscript.
Re: Thank you. We modified the figures and report R2 and p values in the caption of all figures.
Re: We agree that writing “is related” may be misleading. Thus, we changed to “is linked”, which might be less confusing (see page 11, line 203ff). Additionally we agree that this topic is complex, and that there are more factors possible responsible for an acute hypoxia HRmax decline. This is nicely summarized by Mourot (Limitation of Maximal Heart Rate in Hypoxia: Mechanisms and Clinical Importance), who specifically addressed acute hypoxia exposure too. The autonomous nervous system seemed a good candidate (“It has been hypothesized that an increase in parasympathetic activity, rather than decreased sympathetic activity, might be responsible for the blunting of the cardiac chronotropic function during acute hypoxia (Lundby et al., 2001a).”), yet to show a more complete picture we also include other factors. Moreover, we changed the sentence “A higher SpO2 might grant higher HRmax and such a higher performance level” to “A higher SpO2 might be related to a higher performance level” in order to be more careful (see page 11, line 209).
Re: We agree and changed to “reasonable number” (see page 12, line 262).
Re: All running tests have been performed on a treadmill, where air density is negligible. A comparison between HH and NH in regard to cycle and treadmill exercise has not been performed before and thus in our view may have some value. Of course, we recognize and respect the reviewer point of view.
Re: Thank you very much, we inserted these papers as ref. #40 and #55.
Thank you very much again!
Reviewer 2 Report
A focused review on the maximal exercise responses in hypo- and normobaric hypoxia: Divergent oxygen uptake and ventilation limitations.
This study aimed to investigate a very relevant, and applicable, topic – are there differences in key maximal cardiorespiratory measures and VO2max between normobaric and hypobaric hypoxia during cycling vs. running incremental exercise. The Authors have done a nice job outlining these differences, as found in existing literature, and suggesting potential mechanism to explain the varied responses between NH and HH.
While I appreciate the relatively short manuscript, at times a more comprehensive review/comparison to existing literature would strengthen the manuscript.
Some general and specific comments for the authors to consider:
- At times, the authors use the word “simulated”; however, it is not clear, throughout the manuscript, whether 1) only studies that utilized simulated conditions were selected, 2) simulated hypoxia was only used for NH, but not HH, or 3) simulated hypoxia was used for both NH and HH but not in all studies. Please make this more clear throughout the manuscript, as this would also determine whether it would be more appropriate to use the phrases “simulated altitude” and “hypoxia” rather than “altitude”. Additionally, if simulated hypoxia studies are included, please state how many of those studies were blinded and discuss possible a placebo effect.
Title
- If I understand correctly, this study did not assess cardiorespiratory limitations, nor do any of the parameters reported indicated VO2 or VE limitations, per se. Please consider changing the word “limitations” to “responses”, or even “limits” which does not imply that limitation to maximal exercise differ.
Abstract:
- Add “differently” after affected. It is well known that cardiorespiratory outcomes are affected by hypoxia. What is unknown is whether normo- and hypobaric hypoxia affect these outcomes to a different extent.
Intro:
- Please provide a more recent reference instead/in addition to (3)
- Please make it clearer, from the beginning, that these statements refer to acute hypoxic exposure (provide references that relate to chronic responses as well)
- Study aim – Simulated, terrestrial or both NH/HH?
Methods:
- How many of the studies were blinded/ randomized?
- Line 73-74 – again, were both simulated and terrestrial HH studies included? How many studies in each category?
- Add effect size classification – categories for small, moderate, etc ES, and a reference
- Line 81 – what linear regression analyses were performed?
Results:
- It appears that throughout the manuscript there is a misuse of the word “low” with regards to altitude. If “low” refers to altitudes of 2,182m, change to “moderate”, if it refers to simulated altitudes change to “sea level” in Line 85, and throughout the manuscript.
- Line 87 – add “Baseline… exercise testing at SEA LEVEL and altitude”
- Line 90 – Table 2 or 1?
- It strikes me surprising that only 1 study in the last decade investigated responses to incremental exercise in hypoxia.
- Table 2 –
- Title: why “limitations” after “oxygen uptake and ventilation”? (same comment as for the title of the manuscript)
- The first conclusion is very general and not clear – again, why “limitations”?
- 5th bullet point is not clear. What do you mean by “performance limitations threshold”?
- To make Table(s) 3 more clear/easier to interpret by the reader, rather than presenting the absolute values consider reporting the % change from sea level to altitude (in the bottom part of each table) for all (combined data), cycling, and treadmill. As it appears, these tables are hard to follow and interpret, and %difference will cut down 3 columns and allow the reader to compare differences between exercise tasks within a condition (NH or HH).
- Line 116 – for both cycling and treadmill combined?
- Line 122 – the conclusion regarding NH cannot be stated if no ES are presented (and thus a comparison is not possible)
- Line 124-126 – please restate this sentence, as written, it reads as if the ∆VO2max was different for a given ∆SpO2, which was not the case (otherwise the regression lines would not be parallel)
- Table 4 – in the caption below please indicated 1) what the 2 values under the “n” column refer to, 2) why ES was not calculated for all NH parameters
- Figures 2 & 3
- Add legends.
- Were these relationships significant? Need to indicate this in the text. Also suggest adding p and r values on the figure (not in caption)
- P and r values should at least be reported in text
- In the figure caption
- Although it may be obvious, please consider adding how the delta was calculated (e.g. altitude – sea level or vice versa) either in text or axes labels.
Discussion
- Line 142-143 – please add whether these findings pertain to both conditions combined and or HH (unless they are also true for NH, it which case these results should be presented above in the results section).
- Line 149-151 – I believe this claim is not entirely correct. Clear differences in in VE, SpO2 and NO have been shown between normo- and hypobaric hypoxia; Suggest reading the recent Crosstalk review by Millet and Debevec in J Phys (2020).
- Line 155 (and throughout) – please indicate that differences were found between acute HH and NH, as these findings are not necessarily true for chronic exposure.
- Line 165 – why “required”? VEmax represents the “supplied” VE, not the demanded VE (which, in the case of altitude/hypoxia and specifically in trained individuals, demand could > supply). Is it possible that because subjects/athletes are able to breathe more (higher VE) in HH, VO2max is impaired to a lesser extent compared to NH?
- Line 167 - Please provide a reference as to why differences in VEmax, per se, would affect pH. The whole discussion regarding pH is very speculative and unrelated, please consider removing.
- Paragraph starting in line 177
- From the results of the current study, a cause and effect relationship between ∆SpO2 and ∆VO2max cannot be determined, and therefore it cannot be stated that a “decrease in SpO2… reduced VO2max”.
- As mentioned above, please report the relationship between ∆HRmax and ∆SpO2max for both HH and NH in the results, with p values.
- Line 182 – in theory, this claim is correct for a given VE value. However, it may be irrelevant for the current analyses since VEmax was higher in HH, and therefore it cannot be concluded or assumed that respiratory muscles required less energy at maximal exercise.
- It is important to recognize that 1) SpO2max and HRmax values were obtained at different exercise intensities (given that VO2max was different between NH and HH), and 2) cause and effect cannot be concluded from correlations, and therefore any relationship/interaction between SpO2, HRmax, VO2max and VE max should be interpreted with cation.
- While I appreciate the short and concise discussion, please include a more comprehensive discussion of your results concerning differences between cycling and running and reference the appropriate literature.
- Lines 215-216 – “low and acute simulated (normobaric and hypobaric) high altitude” – again, this is very vague and not consistent with earlier wording/description of the conditions tested:
- What does the “low” refer to? The lowest “altitude” included in the analysis was 2182m, which is considered as “moderate” altitude
- Line 208 – this limitation is very vague, please be more specific
Author Response
Re: Dear Reviewer: thank you very much for all your efforts in order to improve our manuscript. Your comments are really helpful and constructive. We tried to respond adequately to all of them. Please, see the point-to-point answers below.
Re: Thank you for this important point. We now indicate with HH whether the study was performed in a hypobaric chamber (HC) or at terrestrial altitude (TA) (table 2, see page 4f). There is only one study that was performed at RA (Horstman et al. 1980). We did not exclude this study because it may be interesting for comparison reasons. It did not change the presented findings when excluded. To better explain, we now write “… evaluated at sea level and altitude (i.e., in NH or HH corresponding to altitude levels varying between 2,182 and 5,400 m)”. Furthermore, we state “… were performed in HH (all in hypobaric chamber except one which was performed at real altitude)… ” (see page 4, line 92ff). Unfortunately, available information does not allow to assess with confidence whether participants were blinded in NH or not. This has been added to the limitation section also referring to a potential placebo effect (see page 12, line 258ff).
Re: Thank you, agreed. We changed the title as suggested.
Re: Yes, done as suggested (see page 1, line 22).
Re: As we refer to historical studies, more recent references are probably not appropriate. However, recent studies have been cited later in the text.
Re: Yes, we now made this clear by adding “acute”. We also refer to subacute/chronic exposure as follows: “It has to be noted that during the acclimatization process (with subacute and chronic exposures to altitude/hypoxia) physiological responses change associated with improved exercise performance”.
Re: Thank you. We added “HH: simulated and terrestrial” (see page 2, line 64).
Re: As mentioned above this cannot be assessed with confidence, this has been added to the limitations section (see page 12, line 258ff).
Re: We added “HH: simulated and terrestrial” (page 2, line 64).
Re: An ES below 0.5 was considered as small, 0.5 to 0.8 as medium and > 0.8 as large as suggested by Cohen (1988) (page 4, line 88f)
Re: Simple regression analyses. This information has been added (page 2, line 86).
Re: Thank you for this point. Yes, “low altitude” has been changed to “sea level” throughout the manuscript.
Re: Done as suggested.
Re: Sorry for this mistake. Now correctly numbered as Table 1.
Re: Yes, we agree, but based on our search strategy, we did not find any additional appropriate study.
Re: Has been changed as suggested.
Re: Limitations has been replaced by “responses” and performance limitations by “performance reduction”. Finally, as suggested by reviewer 3, we replaced the table by a “didactic figure” (figure 2):
Re: Oh yes, we discussed this problem extensively when preparing the manuscript and now again. Finally, we decided to remove the absolute values in former table 3 (now table 2, page 6) and only to present characteristics there. Percentage differences (per 1 km gain in altitude to provide reasonable comparisons) are presented in former table 4 (now table 3, page 8). This seemed important to us due to the different “altitudes”. We hope you can agree. Thank you for understanding.
Re: Yes, first both exercise modes then separately; has now been indicated
Re: Right, has been modified (page 6, line 108f).
Re: Thank you, right. The first part of this sentence has been deleted (page 9, line 137ff).
Re: Numbers have been indicated; ES for treadmill running in NH could not be calculated because only one study was available.
Re: Thank you. The legends of former Figures 2 and 3 (now Figures 3 and 4) have been added and R2 and p-values (both are < 0.05) are reported in the figure caption (page 9f).
Re: We agree. The following information has been added: “Delta values are calculated as HH/NH values minus sea level values”.
Re: We stated: HH (simulated and terrestrial) (page 10, line 165f).
Re: We stated that more carefully and included the suggested references as #40 and #55 (page 10, line 174f).
Re: Done as suggested (page 10, line 181).
Re. Yes, fully agreed. “Required” has been deleted.
Re: Agreed, this part has been removed as suggested (page 11, line 193f).
Re: we changed to “was related to” (page 11, line 203).
Re: Yes, done as suggested.
Re: Yes, right. This sentence has been removed (page 11, line 208).
Re: Yes, has now been discussed more carefully (page 11, line 204ff).
Re: Not to become to extensive, we included results from the review presented by Millet and colleagues (2009) and expanded the discussion section (page 11, line 219ff).
Re: Thank you. Has been changed (throughout the paper) according to the suggestions above.
Re: The limitation section has been modified and expanded also in response to the comments of the other reviewers.
Thank you very much again for the really rigorous review.
Reviewer 3 Report
In this study, the author reviews the cardiovascular responses to maximal exercises performed in hypo and normobaric hypoxia. The focus of the paper is well phrased, and this review is well written. To my knowledge, it is also the first review that aims to clarify this topic. The calculation of effect size was also important to reinforce the analyses. However, there are some issues with the paper that should be clarified. I have listed some specific questions in this regard below.
Comments:
- To my opinion, the authors need to specify that the study focusses on the "acute" responses in the introduction and in the abstract.
- Why did the authors exclude athletes with VO2max lower than 50ml/min/kg?
- Maybe the authors should indicate somewhere that this review focussed on moderate trained athletes (maybe because there is no study performed in well trained athletes). In the same way, altitude was "moderate" or "high".
- Is there limitation in the literature about lower altitude level or no study was conducted at an altitude lower than 2182m? If there is some data at lower altitude, it will be relevant for the reader to have some indications (for a practical point of view). Indeed, there are several elite sport structures in the world at only 1600-2200m.
- In my opinion, some characteristic of the subjects needs to be included in the table or somewhere, especially if athletes are acclimated or not and the time spent at altitude before the studies.
- Table 2: Maybe the key findings should to be redacted and resumed in a didactic figure. Please see the following point.
- I suggest including a didactic illustration (figure) to resume the different responses between HH and NH.
- Tables 3 and 4: is it possible that some differences may be a consequence to the material used to measure cardiorespiratory responses (K4B2 versus MEATAMAX; SpO2 analyses, etc.)? I think this is a limitation.
- Throughout the manuscript: by "a focussed literature search, do the authors mean "a focussed literature review" (or analysis)?
- I feel that the authors should also present future directions that have to be prospected in this field. For example, practical recommendations for athletes and coaches or other (future directions). I think it will be relevant for the reader to quickly describe the benefices of training in hypoxia in the discussion, especially if the authors are able to do a link between these findings and recommendations on exercise training.
Author Response
Dear Reviewer: thank you very much for your helpful and constructive comments!
Re: Fully agreed. As also suggested by the other reviewers we now emphasized this important point and added “acute” wherever appropriate.
Re: We intended to focus only on well-trained athletes because those subjects show more pronounced reduction in VO2max as demonstrated by Ferretti and colleagues (1997). This is now pointed out in the inclusion criteria (see page 3, line 75ff).
Re: Thank you. Yes, we now stated that we intended to focus on well-trained athletes and on all available studies comparing acute maximal exercise responses between sea level and acute altitudes (above 2000 m).
Re: Yes, you are right. We have chosen altitudes above 2000 m because physiological responses and performance reductions are rather small below 2000 m.
Re: Thank you. We only can additionally state that subjects were not exposed to altitudes above 2000 m for at least one month prior to the study.
Re: We agree. We tried to present such an illustration as Figure 2.
Re: Oh yes, thank you for this point that cannot be excluded and has now been added to the limitations as follows: “Finally, potential confounding may arise from the use of different test protocols and different measurement devices” (page 12, line 261f).
Re: Yes, you are right. We now use the wording “a focused literature review”.
Re: thank you for this important point. We added to the conclusions: “Future directions: Future studies will focus on potential sex and age differences between responses to acute NH and HH but will also evaluate changes occurring with acclimatization” (page 12, line 275).
Thank you very much again!
Round 2
Reviewer 1 Report
The authors have proposed substantial improvements in their manuscript.
Author Response
Thank you very much again!
Reviewer 2 Report
Thank you for responding to my comments; almost all have been addressed and the manuscript reads and flows much better now. A few changes, however, have not been made even though the authors state that they have.
Below please are my comments:
Figure 2 is GREAT! Two minor suggestions:
1) Add in the figure legend what 1, 2 or 3 arrows indicate
2) Take out the effect sizes in Table 3 as these data are now presented in the figure - this will make Table 3 less cluttered and more readable (you could add superscripts to indicate small/moderate/large ES)
* Line 116 – for both cycling and treadmill combined?
Re: Yes, first both exercise modes then separately; has now been indicated.
I don’t see that this was added in the revision
*Line 122 – the conclusion regarding NH cannot be stated if no ES are presented (and thus a comparison is not possible)
Re: Right, has been modified (page 6, line 108f).
This has not been addressed in the revised manuscript
* Table 4 – in the caption below please indicated 1) what the 2 values under the “n” column refer to, 2) why ES was not calculated for all NH parameters
Re: Numbers have been indicated; ES for treadmill running in NH could not be calculated because only one study was available.
This was not addressed in the revised manuscript
Figures 2 & 3
Add legends.
Were these relationships significant? Need to indicate this in the text. Also suggest adding p and r values on the figure (not in caption)
P and r values should at least be reported in text
In the figure caption
Re: Thank you. The legends of former Figures 2 and 3 (now Figures 3 and 4) have been added and R2 and p-values (both are < 0.05) are reported in the figure caption (page 9f).
I don’t see legends, or p values on figures 3 & 4
Discussion
Line 142-143 – please add whether these findings pertain to both conditions combined and or HH (unless they are also true for NH, it which case these results should be presented above in the results section).
Re: We stated: HH (simulated and terrestrial) (page 10, line 165f).
This was only stated in the previous sentence
Line 149-151 – I believe this claim is not entirely correct. Clear differences in in VE, SpO2 and NO have been shown between normo- and hypobaric hypoxia; Suggest reading the recent Crosstalk review by Millet and Debevec in J Phys (2020).
Re: We stated that more carefully and included the suggested references as #40 and #55 (page 10, line 174f).
Can’t say I agree with the wording of this sentence, as most studies do not report “conflicting” findings with regards to VE, SpO2 and NO
As mentioned above, please report the relationship between ΔHRmax and ΔSpO2max for both HH and NH in the results, with p values.
Re: Yes, done as suggested.
Still missing
Author Response
Re: Sorry for that unintended errors which are partly due to different versions of revised manuscripts.
Re: thank you, 1) legend has been extended according to your suggestion and 2) table 3 has been modified as suggested; effect sizes have been indicated by letters a, b, and c.
Re: Yes, first both exercise modes then separately; has now been indicated.
Re: Sorry, we now state “combined for cycling and treadmill running”
Re: Right, has been modified (page 6, line 108f).
Re: Sorry again; the conclusion regarding NH has now been deleted.
Re: Numbers have been indicated; ES for treadmill running in NH could not be calculated because only one study was available.
Re: Sorry again; we now explained the meaning of those numbers and also stated “ES for treadmill running in NH could not be calculated because only one study was available”.
Re: Thank you. The legends of former Figures 2 and 3 (now Figures 3 and 4) have been added and R2 and p-values (both are < 0.05) are reported in the figure caption (page 9f).
Re: The legends for the figures are highlighted (yellow) and p-values have been included on the figures and the figure caption.
In addition, description of these findings (fig. 3 + 4) was not correct and has been changed (please, see below).
Re: We stated: HH (simulated and terrestrial) (page 10, line 165f).
Re: Thank you. We stated now as follows: “Furthermore, during treadmill running in HH (simulated and terrestrial) when compared to cycling, a larger SpO2max and HRmax decrease per 1,000 meters of altitude gain was found.”
Re: We stated that more carefully and included the suggested references as #40 and #55 (page 10, line 174f).
Re: We agree; this sentence has now been deleted.
Re: Yes, done as suggested.
Re: Oh yes, our wording with regard to fig. 3 and 4 was wrong. We modified as follows, “Besides, data show a similar slope between the changes in VO2max and SpO2max (Figure 3) and VEmax and HRmax (Figure 4) with gain in altitude, but at different levels for NH and HH.
Thank you very much again!
Round 3
Reviewer 2 Report
With regards to Figure 2 - The description of the arrows in the figure legend is not satisfactory for a scientific paper. What do you mean by “more increase than XX”? Is that within a variable or a certain %change? What objective criteria were used to determine whether to put 1, 2 or 3 arrows?
Please fix.
Author Response
Thank you for this again valuable input. We have revised Figure 2 and the corresponding legend.